# Epidemiologic Study of Gene Distribution in Romanian and Brazilian Patients with Non-Cicatricial Alopecia

**DOI:** 10.3390/medicina59091654

**Published:** 2023-09-13

**Authors:** Monica Păun, Gustavo Torres, George Sorin Țiplica, Victor Mihail Cauni

**Affiliations:** 1Dermatology Research Unit, Colentina Clinical Hospital, 020125 Bucharest, Romania; 2Department of Dermatology, Carol Davila University of Medicine and Pharmacy, 020021 Bucharest, Romania; george.tiplica@umfcd.ro; 3Human Genome and Stem Cell Research Center, University of São Paulo, São Paulo 05508-220, Brazil; gustavotsouza@hotmail.com; 4Second Department of Dermatology, Colentina Clinical Hospital, 020125 Bucharest, Romania; 5Department of Urology, Colentina Clinical Hospital, 020125 Bucharest, Romania; victorcauni@yahoo.com

**Keywords:** alopecia areata, androgenetic alopecia, genetic test, DNA analysis

## Abstract

*Background*: Androgenetic alopecia (AGA) and alopecia areata (AA) are the most common types of non-cicatricial alopecia. Both diseases have limited effective therapeutic options and affect patient quality of life. Pharmacogenetic tests can help predict the most appropriate treatment option by evaluating the single nucleotide polymorphisms (SNPs) corresponding to genes related to alopecia. The objective of the study was to evaluate and compare selected SNPs and genes in AA and AGA patients from Romania and Brazil. *Materials and Methods*: We performed a retrospective study regarding the associations between AA and AGA and 45 tag SNPs of 15 genes in 287 Romanian and 882 Brazilian patients. The DNA samples were collected from oral mucosa using a swab. The SNPs were determined by the qPCR technique. Each genetic test displays the subject’s genotype of the selected gene and the prediction of a successful treatment (e.g., genotype AA of the GR-alpha gene is related to a predisposition to normal sensibility to topical glucocorticoid, and, therefore, glucocorticoids should be effective). *Results*: The *GR-alpha*, *GPR44-2*, *SULT1A1*, and *CRABP2* genes were statistically significantly different in Brazil compared to Romania. The *SULT1A1* activity that predicts the response to minoxidil treatment showed in our analysis that minoxidil is recommended in half of the cases of AGA and AA. Patients with AGA and a high expression of *SRD5A1* or *PTGFR*-2 may benefit from Dutasteride or Latanoprost treatment, respectively. Most of the studied genes showed no differences between the two populations. *Conclusions*: The DNA analysis of the patients with alopecia may contribute to a successful treatment.

## 1. Background

Androgenetic alopecia (AGA) and alopecia areata (AA) are among the most common types of non-cicatricial alopecia. AGA is characterized by a distinctively receding frontal hairline in men (Figure 1) and diffuse hair thinning with retention of the frontal hairline in women. Patients with AGA have a shorter anagen phase of the hair cycle, which will gradually diminish the hair follicle (HF) and cause the development of vellus hairs [1]. AGA’s etiology and pathophysiology are not entirely understood; however, genetics [2] and androgens are implicated [3]. AA is a complex autoimmune disease with a genetic substrate that targets the HF in the anagen phase [4]. It typically presents with round alopecic patches (Figure 2) and can affect children and adults. AA [5] and AGA [6] evaluation and diagnostic should include a medical history, clinical examination, dermoscopy, and hair pull test. Additional investigations for AA, such as scalp biopsy, are usually unnecessary [5]. The effectiveness of most treatments for AGA and AA is variable. Consequently, utilizing DNA variants may have an impact on treatment strategies.

The current analysis evaluates and compares the associations between non-cicatricial types of alopecia (AGA and AA) and 45 tag single nucleotide polymorphisms (SNPs) and 15 genes identified from previous AA and AGA association studies [7,8,9,10,11,12,13,14,15,16]. The evaluated population comprises 287 patients from Romania and 882 from Brazil, clinically diagnosed with AA or AGA.

Local treatments of AGA are limited and usually consist of minoxidil (2% BID or 5% QID), 5 alpha-reductase (finasteride and dutasteride), and antiandrogens [6]. Regarding AA, topical and intralesional corticosteroids remain the first-line treatment. Other topical treatments, such as minoxidil and prostaglandin F2α analogues, are also prescribed in AA [5].

## 2. The Objective of the Study

The study aims to evaluate and compare genetic data (selected SNPs and genes) in patients from Romania and Brazil diagnosed with AA and AGA.

## 3. Materials and Methods

We performed a retrospective study regarding associations between AA and AGA and 45 tag single nucleotide polymorphisms (SNPs) of 15 genes in a total of 1169 patients: 287 Romanian patients (group 1) and 882 Brazilian patients (group 2). All the patients of both groups were diagnosed with AA or AGA. The diagnosis of AGA or AA was confirmed by healthcare professionals in private clinics from different cities in Romania and Brazil, based on the evolution of the disease, the clinical aspect, and the trichoscopic examination. All patients signed the informed consent. The DNA samples were collected from oral mucosa using a swab. The sampling method was non-invasive and ensured adequate DNA quantity and quality. All the genotyping assays were performed by qPCR Taqman Assays (Thermo Fisher Scientific, USA) from the genomic DNA collected from the patient.

Fagron Genomics (Barcelona, Spain) provided all anonymized data from their database for the current retrospective study.

The evaluated SNPs and the related genes were rs9282861(*SULT1A1* gene), rs523349 (*SRD5A2* gene), rs39848 (*SRD5A1* gene), rs10782665 (*PTGFR-3* gene), rs1328441 (*PTGFR*-*2* gene), rs6686438 (*PTGFR-1* gene), rs13283456 (*PTGES2* gene), rs2229765 (*IGF1R* gene), rs6198 (*GR-alpha* gene), rs533116 (*GPR44-2* gene), rs545659 (*GPR44-1* gene), rs2470152 (*CYP19A1* gene), rs12724719 (*CRABP2* gene), rs13078881 (*BTD* gene), and rs4343 (*ACE* gene). The study was approved by the Bioethics Committee of Colentina Clinical Hospital, Bucharest, Romania (approval no. 17/22.12.2022).

### 3.1. Statistical Analysis

Data were collected, analyzed, coded, and introduced in the Statistical Package for Social Science (IBM SPSS, New York, NY, USA) version 23. Quantitative data were presented as several patients from each country and percentages of the patients for each SNP in each alopecic disease (AGA and AA), as illustrated in Table 1.

The Chi-square test provided information about the relationship between the variables. The test was based on the observed frequencies fi, which represented the frequencies of occurrence of the component elements of each SNP and country. The SNP variable component elements were the country variable genotypes (Brazil and Romania). Based on the observed frequencies, the theoretical frequency for each observed frequency was determined according to a specific calculation algorithm. Each cell with observed frequencies (fi) corresponded to a cell with theoretical values (fti). Depending on the determined Chi-square value (χ^2^) and the degree of freedom (df), the Chi-square test provided the actual significance threshold, denoted by *p*, which was compared with the significance threshold considered in statistical research, in our case α = 0.05. The decision regarding the association (dependence) or non-association (independence) of the two variables was taken depending on the comparison between the thresholds *p* and α.

The tested research hypotheses were the null (H0) and alternative hypothesis (H1). The null hypothesis stood for no association between an SNP variable and its genotypes and the country variable (Brazil and Romania), with the variables being independent. The alternative hypothesis (H1) referred to an association between an SNP variable and its genotypes and the country variable (Brazil and Romania), with the variables being dependent. If *p* ≤ 0.05 (α), the null hypothesis was rejected, and there was a relationship between the two variables in the sense that the data (genotypes) of each SNP depended on the country. If *p* > 0.05 (α), the null hypothesis was accepted, and there was no connection between the two variables in the sense that the genotypes of each SNP did not depend on the country (they were similar).

### 3.2. Principal Component Analysis

Principal component analysis (PCA) is a statistical technique used by condensing the dimensions to the few principal components (PCs) that best describe the main patterns. It is used for analyzing data containing a large number of observations [17].

We performed a PCA for our 1169 patients with AA and AGA. The internal consistency coefficient (Cronbach’s Alpha) of the two specified dimensions and the percentage values for the variance on each dimension and for the total (18.142%) is displayed in Table 2.

The Variance Accounted For table (Table 3) shows the coordinates of each element on each dimension relative to the centroid (0, 0). Furthermore, features with a very small mean coordinate indicate that these elements do not contribute substantially to the principal components. The SNPs rs39848 (*SRD5A1* gene), rs523349 (*SRD5A2* gene), rs12724719 (*CRABP2* gene), and rs13078881 (*BTD* gene) have mean values less than or equal to 0.010, meaning they do not adequately contribute to the principal components. After transforming the variables through the optimal analysis approach, the Correlation-Transformed Variables table (Table 4) shows the correlations between the components. We observed that the elements contributing substantially to the main features have an eigenvalue > 1.

The Component-Transformed Variables table (Table 4) indicates the coordinates for each SNP on each dimension. Based on the data, the plotting of the analyzed SNPs according to the coordinates agree with to the two dimensions. We observed that the elements that contribute substantially to the main components are those with an eigenvalue > 1.

## 4. Results

In group 1 (287 Romanian patients), 90.2% were diagnosed with AGA and 9.8% with AA. In group 2 (882 Brazilian patients), 94.2% were diagnosed with AGA and 5.8% with AA.

Four SNPs (rs6198, rs533116, rs9282861, and rs 2724719) with the related genotypes (*GPR44-2, GR-ALPHA, SULT1A1*, and *CRABP2*) were statistically significantly associated in Brazil and Romania (Table 5). In these cases, the null hypothesis was rejected, with the data being significant from one population to the other. The non-statistically significant SNPs are shown in Table 6.

The most prevalent genotype (GA) was observed in the following SPNs: rs533116 (*GPR44-2* gene), rs2470152 (*CYP19A1* gene), rs1328441(*PTGFR-2* gene), rs4343 (*ACE* gene), and rs2229765 (*IGF1R* gene). Prostaglandin D2 receptor 2 (*GPR44* or *CRTH2*) variants were associated with an increased *GPR44*, resulting in higher responsiveness to prostaglandin D2 and HF regression. The aromatase gene (*CYP19A1*) showed low testosterone conversion in estrogens and high conversion into DHT. Prostaglandin F receptor 2 (*PTGFR-2*) was mainly related to the treatment efficacy with prostaglandin analogue and latanoprost. The angiotensin-converting enzyme (*ACE*) variants were associated with increased plasma levels of angiotensin in 49.0% of the patients with AA from Brazil and 54.8% from Romania, and insulin-like growth factor-I (*IGF-I*) variants were associated with lower plasma *IGF-1* levels in half of the evaluated patients.

The genotype AA was more frequent for the tag SNP rs6198 (*GR-ALPHA* gene) and rs545659 (*GPR44-1* gene) in both countries. The SNP rs6198 analysis indicated that the glucocorticoid receptor (*GR* or *NR3C1*) variants are associated with resistance or sensitivity to corticosteroids. Regarding rs545659 SPN, most subjects have an increased *GPR44*, leading to higher responsiveness to prostaglandin D2 and HF regression. Our analysis showed tha the AA genotype of the *GPR44-1* gene was found in 68.6% of the cases from Brazil, 63.1% in AA, 75.0% of AGA cases from Romania, and 62.9% of AA cases.

For the tag SPNs rs9282861(*SULT1A1*), rs523349 (*SRD5A2* gene), and rs13283456 (*PTGES2* gene), the CC genotype was more prevalent. In our study, the treatment with minoxidil at a regular concentration showed good results in more than half of the patients with AGA (52.9% in Brazil and 50.0% in Romania) and AA (52.6% in Brazil and 53.3% in Romania). The *SULT1A1* gene can predict a good response to minoxidil treatment.

The GG genotype was most frequently associated with the following SPNs: rs12724719 (*CRABP2* gene), rs13078881 (*BTD* gene), and rs6686438 (*PTGFR-1* gene). The SNP analysis did not indicate the necessity to supplement with retinoic acid or vitamin B as the levels were physiological. The genetic testing showed that latanoprost might not respond positively in 60.8% of AGA in Brazilian patients and 46.4% of Romanian patients.

The gene *PTGFR-3* SPN rs10782665 was commonly associated with genotype GT related to latanoprost treatment efficacy. Regarding the SPN rs39848 (*SRD5A1* gene), the more common genotype was CT. The genetic analysis showed that increased *SRD5A1* activity leads to increased DHT levels and hair growth inhibition, and treatment with Dutasteride is recommended in 43.1% of Brazilian and 53.6% of Romanian AGA patients. Figure 3 plots the *p*-value for each SPN. It is confirmed that the four values greater than 1.30 correspond to the SNPs seen in Table 5. The lowest nominal *p*-value was observed for the tag SNP RS9282861 gene *SULTA1A* (*p* = 0.001).

Regarding the type of alopecia (AA or AGA), the Chi-square test highlighted a significance threshold of *p* = 0.02 < 0.05. This result shows that AGA and AA are associated with the country variable, meaning that these types significantly differ from country to country, and the null hypothesis was rejected.

Based on the PCA data, the components plotted according to the coordinates corresponding to the two dimensions are shown in the Figure 4. SNPs rs523349, rs13078881, rs39848, rs12724719, and rs13283456 had means below 0.010, being localized around the coordinate (0, 0).

## 5. Discussion

The possibility of linkages between SNPs in AA and AGA has only recently been investigated in a small number of Australian [7], European (Polish [5], UK, and German [8]), Chinese and European [9], and Korean [10] populations.

### 5.1. Aromatase Gene (CYP19A1 Gene)

Aromatase inhibitors (AIs), which inhibit the synthesis of estrogens, cause a relative increase in the activity of 5-reductase. Aromatase is known to reduce intrafollicular testosterone [5] and DHT levels [2,6] and to catalyze the conversion of testosterone to estradiol. This increase in the aromatase can determine “pseudo male pattern androgenetic alopecia,” which results in male pattern hair loss that mimics female androgenetic alopecia (FAGA) [11]. Aromatase expression varies between alopecic and non-alopecic scalps [18]. According to Yip et al., the aromatase gene (*CYP19A1*) may predispose women to types of non-scarring alopecia [7]. Estrogens can both prevent and modulate hair growth; hence, *CYP19A1* plays a crucial protective role in frontal hair lines [7]. Compared to men, women have more significant levels of *CYP19A1* in their frontal and occipital follicles [19].

Fifteen menopausal women with hormone receptor-positive breast cancer who received AIs were the subjects of a study by Rossi et al. [11]. After receiving AI therapy for a year, the patients began to develop frontotemporal follicle miniaturization that mimicked FAGA. Yip et al. [7] performed a gene-wide study of the association between female pattern hair loss (FPHL) and the aromatase gene-encoding *CYP19A1*. The investigators examined 61 *CYP19A1* SNPs in the Australian population of 484 Caucasian women with FPHL and 471 controls. The CC genotype of rs4646 was substantially more common in FPHL (58%) than in the controls (48%), indicating a recessive genetic influence, even though no significant relationship was established at the allelic levels. Even though our study did not analyze the sex of the patients with AGA, the most prevalent genotype of the *CYP19A1* gene (SPN rs2470152) was GA (47.4% in Brazil and 47.7% in Romania), showing a predisposition to reduced *CYP19A1* activity.

### 5.2. Angiotensin-Converting Enzyme (ACE Gene)

The immune-dependent disease known as AA is defined by the interaction of T cells with follicular antigens. Studies have revealed the presence of a local renin–angiotensin system in the skin, where the angiotensin-converting enzyme (*ACE*) is involved in autoimmunity and causes alopecia due to HF’s chronic inflammation [20,21]. A few cases of non-cicatricial alopecia induced by *ACE* inhibitors have been identified. Lisinopril-induced alopecia in a patient of Kataria et al. [10] and enalapril-induced alopecia in a patient of Ahmad et al. [22] were reported. Hair re-growth was observed after the cessation of the treatment. A patient of Motel et al. received *ACE* treatment for one month, and after the medication was stopped, hair thinning was observed at a follow-up examination [12].

Namazi et al. assessed the serum activity of *ACE* in a study with 19 patients with AA and 16 healthy control participants. Although the findings were not statistically significant, AA patients had higher serum *ACE* levels. Moreover, in the patient group, there was no association between *ACE* activity and the severity or duration of the condition [13]. Fahim et al. found a significant correlation between serum *ACE* and disease severity despite the lower serum *ACE* levels in AA patients. Angiotensin I, which may also be involved in the inflammation of AA, is considered to be the cause of *ACE* consumption and reduced tissue levels of the enzyme [21]. In our study, 49.0% of the patients with AA from Brazil and 54.8% from Romania expressed the GA genotype, resulting in a predisposition to an increased angiotensin conversion activity.

### 5.3. 5-Alpha-Reductase (SRD5A1 and SRD5A2 Genes)

Genetics plays an essential role in AGA [2]. The role of the 5-alpha-reductase (5AR) enzyme in AGA is implicated in the metabolism of testosterone to dihydrotestosterone (DHT), hence the 5AR inhibitors effect in treating AGA [14]. Finasteride and dutasteride are 5AR inhibitors commonly used in AGA. Dutasteride is an inhibitor of both type I and II 5ARs and is considered an effective treatment for AGA [23,24]. The treatment response to dutasteride varies in each individual; however, it is unknown if the genetic factors contribute to these variations [23]. It was also observed that the response to dutasteride might be related to variations of genes involved in the metabolism of steroid hormones, such as *SRD5A1*. Finasteride inhibits type II 5ARs only, while dutasteride inhibits type I and type II 5ARs simultaneously. Accordingly, a difference in *SRD5A1* could influence how well an individual responds to dutasteride when treated for AGA [24]. Moreover, in the Korean population, the analysis of the polymorphism of *SRD5A1* and *SRD5A2* genes showed that it might not be directly associated with the development of AGA [15]. Limited data are available regarding other populations. Our analysis shows that dutasteride at standard doses is recommended in AGA in 43.1% of Brazilian and 53.6% of Romanian patients and Finasteride is recommended in 51.0% of AGA in Brazilian patients and 42.9% of AGA in Romanian patients.

### 5.4. Glucocorticoid Receptor (GR-Alpha Gene)

Glucocorticoid receptor (GR) is known to be expressed in human hair follicles [25]. Early studies showed significant findings on glucocorticoid biochemical pathways. The GR is elevated in AA patients who are “glucocorticoid-resistant” and unable to bind glucocorticoids, suppressing cellular transcription [18]. Dexamethasone (DEX) is a synthetic glucocorticoid that inhibits the proliferation of human hair dermal papilla (DP) cells. DEX also decreases the expression of growth factors required for hair growth [26]. In addition, DEX promotes the catagen phase of the hair cycle [25]. Corticosteroids are known to be a treatment for AA. In our AA cases, tag SPN rs6198 *GR-alpha* showed that glucocorticoids should be effective in 73.6% of Brazilian and 67.6% of Romanian patients.

### 5.5. Insulin-like Growth Factor-1 (IGF-1 Gene)

The regulation of insulin-like growth factor-1 (IGF-1) by androgens is well established. IGF-1 increases HF development by controlling cellular proliferation, according to in vitro research. In organ cultures, in the absence of IGF-1 or insulin, anagen HF enters the catagen phase [27]. Panchaprateep et al. reported significantly lower levels of IGF-1 and its binding proteins (IGFBP-2 and IGFBP-4) in DP cells obtained from alopecic versus non-alopecic scalps [27]. We report similar results, as more than half of the patients are predisposed to lower IGF-1 levels.

### 5.6. Prostaglandin D, Prostaglandin E, and Prostaglandin F (PTGFR-1, PTGFR-2, PTGFR-3, PTGES2, GPR44-1, and GPR44-2 Genes)

One of the representative lipid mediator groups, which includes prostaglandins (PGs), is the prostanoid family. They have a variety of physiological impacts on controlling inflammatory reactions, cell proliferation, and apoptosis. Prostanoids are classified into five types: PGD2, PGE2, PGF2α, and PGI2. PGD2 binds to prostaglandin D2 receptors (DP1 and DP2) [28], while PGE2 binds to prostaglandin E2 receptors (EP1, EP2, EP3, and EP4) and PGF2α to the prostaglandin F2α receptor (FP) [29].

Colombe et al. investigated the expression patterns of the vital enzymes involved in the metabolism of PGs in human HF. They reported that PGD2, PGE2, and PGF2 metabolism are expressed in human hair follicles (HFs) and might be associated with hair growth and differentiation [30]. Androgens can regulate prostaglandin D2 synthase (*PTGDS*) to produce prostaglandin D2 (PGD2) [31]. It was reported that the expression levels of PGD2 in men with male pattern alopecia were higher than in the controls [31]. Studies in mice and humans [32,33] have revealed that PGD2 inhibits the re-growth of the HF and hair development, pointing to a role for the DP2 receptor in alopecia. Increased androgen levels may induce PGD2 because androgens have been found to stimulate PTGDS [30]. Kang et al. [30] hypothesized that the expression levels of the DP2 receptor affect hair cycle progression because mRNA is highly expressed in the early catagen phase, and strong staining in the outer root sheath, dermal papilla, and hair matrix was seen. They observed that in mice hair growth cycles, DP2 receptor expression is highest when the hair follicle regresses. Prostaglandin F2 (PTGF2) and Prostaglandin E2 (PTGE2) act synergistically in hair follicles to stimulate hair growth and prolong the anagen phase, which competes with PTGD2. According to a study, PTGES, the enzyme that produces PTGE2, is overexpressed in male patients with early-stage AGA [34].

Rafati et al. [33] observed significant hair re-growth in AA during the treatment with a latanoprost 0.005% solution. In our analysis, *PTGFR-2* showed an increased likelihood of having a positive response to latanoprost in AA for 48.4% of patients in Brazil and 48.4% in Romania, similar to *PTGFR-3* with 43.8% of patients in Brazil and 50.2% in Romania. In AGA, there was a 33.3–60.7% range of recommendation for latanoprost at regular doses.

The aromatase gene showed a low conversion of testosterone in estrogens, and a high conversion into DHT and prostaglandin D2 receptor 2 (*GPR44-2* or *CRTH2*) variants were associated with an expressed GPR44 leading to higher responsiveness to prostaglandin D2 and HF regression. The prostaglandin F receptors (PTGFRs) were mainly related to latanoprost treatment efficacy, known as a prostaglandin analog.

More than 67.9% of the analyzed patients had a predisposition to normal PGE2 levels. The statistical analysis observed a predisposition to normal GPR44 levels in 43.4% (*PTGFR-1*) and 63.4% (*PTGFR-2*) of the Brazilian patients and 46.7% (*PTGFR-1*) and 64.1% of the Romanian patients. Our SNP analysis did not indicate any treatment with prostaglandin D2 inhibitors in patients from both countries. SPN rs545659 showed that most subjects have an increased *GPR44*, which leads to higher responsiveness to prostaglandin D2 and hair follicle regression.

### 5.7. Cellular Retinoic Acid-Binding Protein 2 (CRABP2 Gene)

Alopecia areata (AA) is an autoimmune disease affecting the hair follicles in the anagen phase. According to Duncan et al., vitamin A regulates the immune response and the hair cycle to slow the extension of AA [35]. Retinol, a form of vitamin A linked to retinol-binding protein 4 (RBP4) that is converted to retinoic acid by retinal dehydrogenases and transported to the nucleus by *CRABP2*. Retinol can be esterified or eliminated without or when RBP1/CRBP is saturated [35]. It was observed that RA induces AA, which may be attributed to an array of factors, such as dysregulated immunological function [35]. Genetic studies in mice with AA revealed increased synthesis and retinoic acid (RA) levels in the involved genes. In contrast, RA degradation genes were reduced in AA patients compared to the controls. The facts were confirmed by immunohistochemistry in biopsies from patients with AA, and both control mice and AA models and RA levels were also elevated in C3H/HeJ mice with AA.

In our analysis, 66.1% of AA patients from Brazil and 76.4% from Romania did not require vitamin A supplementation. In comparison, 30.8% of patients with AA from Brazil and 22.8% of patients from Romania were predisposed to reduced retinoic acid transport, and vitamin A would be recommended.

### 5.8. Sulfotransferase (SULT1A1 Gene)

Recent and older studies reported a correlation between *SULT1A1* expression in the scalp and the response to minoxidil [16,35]. Topical minoxidil is the only FDA-approved treatment for AGA and the most commonly prescribed topical treatment for female androgenetic alopecia (FAGA) [7]. Minoxidil is a pro-drug that needs to be bio-activated into minoxidil sulfate to stimulate hair growth by vasodilation. Minoxidil sulfotransferase (*SULT1A1*) catalyzes the reaction in the hair follicle [36]. Individual differences in *SULT1A1* expression in the scalp explain the varieties of clinical responses to topical minoxidil [37]. Therefore, dermatologists could achieve long-lasting persistent results from AGA treatment by using the *SULT1A1* activity assay [35]. The *SULT1A1* gene analysis showed that minoxidil at regular concentrations would be highly recommended in 52.9% of AGA cases and 52.6% of AA cases in Brazil and 50% of AGA cases and 53.3% of AA cases from Romania.

### 5.9. Biotinidase (BTD Gene)

Biotinidase deficiency (BTD) can be categorized as primary and secondary. Primary BTD is an autosomal recessive inherited neuro-cutaneous disorder [38], while a common cause of acquired biotin deficiency is associated with raw egg consumption or several treatments (anticonvulsants, isotretinoin, and antibiotics) [39].

According to Georgala et al. [40], biotinidase deficiency can induce alopecia. The study showed that 12 out of 19 patients with significantly decreased biotinidase activity compared to the controls experienced clinical remission of AA after biotin supplementation. Moreover, Patel et al. [39] performed a literature analysis of 10 cases of alopecia—most patients with an inherited biotinidase enzyme deficiency or being treated with valproic acid. Eight patients experienced hair re-growth after biotin (vitamin B7) supplementation.

Our SNP analysis shows no indication of vitamin B supplementation in more than 90% of the cases of AA and AGA from both countries. Most of the evaluated subjects had normal biotinidase activity.

Our study had several limitations. First, more data are needed regarding the severity, timing, personal and family history, and therapeutic options for the disease. Second, the study’s population was entirely Brazilian or Romanian, which may have limited the results’ applicability to a broader community. Also, the number of the studied groups of patients was unequal, and the sample size can influence the research results.

## 6. Conclusions

The *SULT1A1* gene was expressed in more than half of the patients with AA and AGA from both groups but was higher in the Romanian population than in the Brazilian one. Higher responsiveness to prostaglandin D2 and hair follicle regression due to an expressed *GPR44* was found in half of the Romanian subjects with AGA and three quarters of the Brazilian subjects. Most of the studied genes showed no differences between the two populations, suggesting a common genetic background. Furthermore, the *ACE* gene variants were associated with increased plasma levels of angiotensin, and *IGF-I* variants were associated with lower plasma IGF-1 levels in half of the studied patients. Additionally, our analysis does not indicate the necessity to supplement with vitamin B in most AA and AGA cases from both groups.

Additionally, the PCA showed that the *SRD5A1, SRD5A2, PTGES2, CRABP2*, and *BTD* genes do not contribute substantially to the principal components.

Individual genes frequently provide only polygenic conditions, but identifying essential genes may reveal new and significant treatment targets. Our research indicates that AGA and AA are likely to become predicted by DNA with sufficient accuracy to support common practical uses. Further and extensive studies are necessary.

## Figures and Tables

**Figure 1 medicina-59-01654-f001:**
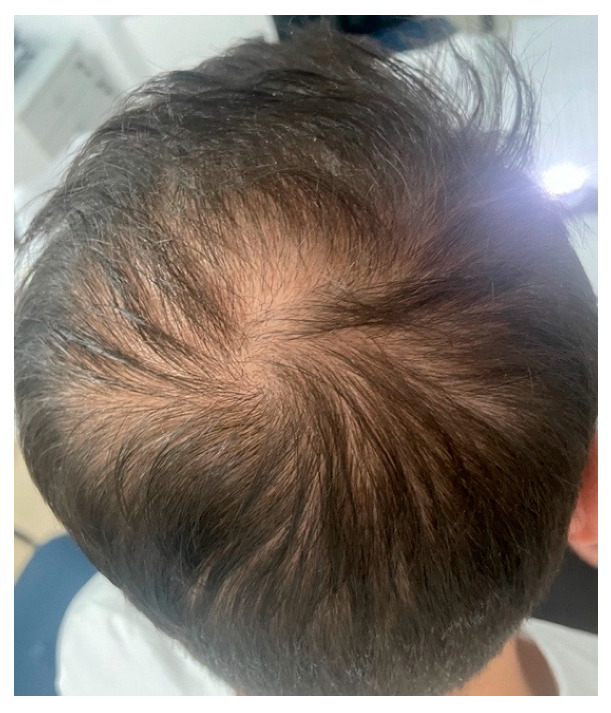
Androgenetic alopecia vertex area. Photograph displayed with patient consent.

**Figure 2 medicina-59-01654-f002:**
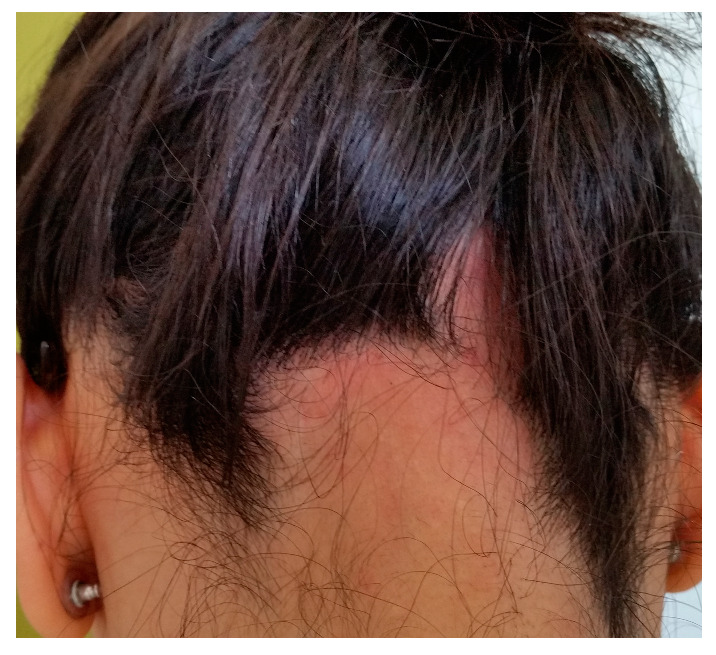
Alopecia areata: round alopecic patches in the occipital area (ophiasis). Photograph displayed with patient consent.

**Figure 3 medicina-59-01654-f003:**
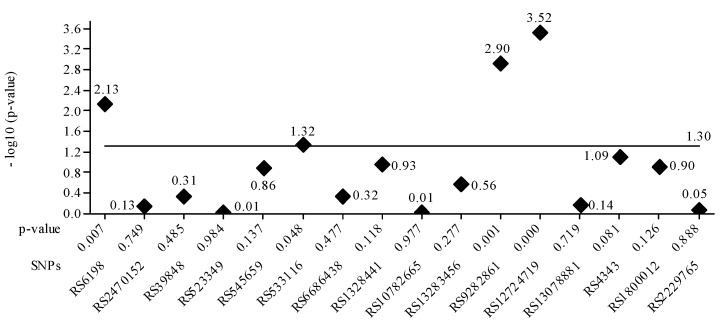
The value of the base-10 logarithm of the *p*-value is always negative because the *p*-value takes values between 0 and 1. For this reason, for the representation of positive values on the Oy axis, the value of the logarithm minus (-log10 (*p*-value)) was used. The significance threshold considered was 0.05 -log10 0.05 = 1.30, the constant value, represented as a horizontal line. Statistically significant values had a *p*-value <= 0.05. For these, the -log10 values are above the line corresponding to *p*-value = 0.05, e.g., above 1.30. It is confirmed that the four values greater than 1.30 correspond to the SNPs in Table 5.

**Figure 4 medicina-59-01654-f004:**
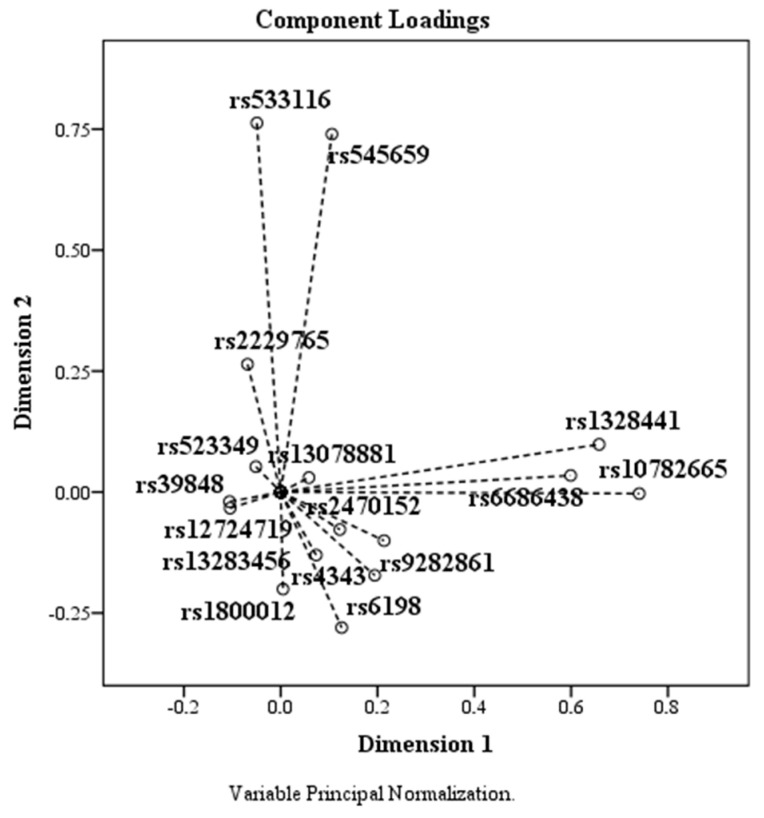
The effect of PCA on SNPs. SNPs rs523349 (*SRD5A2*), rs13078881 (*BTD*), rs39848 (*SRD5A1*), rs12724719 (*CRABP2*), and rs13283456 (*PTGES2*) are located around the coordinate (0, 0).

**Table 1 medicina-59-01654-t001:** Comparation between SNPs in patients with androgenetic alopecia and alopecia areata from Romania and Brazil.

GENA—SNP/Genotype	Brasil (BR), n = 882				Romania (RO), n = 287				Chi-Square Test	
	N	Genotype in BR	AGA	AA	N	Genotype in RO	AGA	AA	df	*p*
GR-alpha-RS6198										
AA	651	73.8%	76.5%	73.6%	189	65.9%	50.0%	67.6%	2	0.007
GA	208	23.6%	19.6%	23.8%	82	28.6%	46.4%	26.6%		
GG	23	2.6%	3.9%	2.5%	16	5.6%	3.6%	5.8%		
CYP19A1-RS2470152										
AA	259	29.4%	29.4%	29.4%	89	31.0%	21.4%	32.0%	2	0.749
AG	418	47.4%	37.3%	48.0%	137	47.7%	42.9%	48.3%		
GG	205	23.2%	33.3%	22.6%	61	21.3%	35.7%	19.7%		
SRD5A1-RS39848										
CC	177	20.1%	25.5%	19.7%	52	18.1%	10.7%	18.9%	2	0.485
CT	418	47.4%	43.1%	47.7%	131	45.6%	53.6%	44.8%		
TT	287	32.5%	31.4%	32.6%	104	36.2%	35.7%	36.3%		
SRD5A2-RS523349										
CC	425	48.2%	37.3%	48.9%	138	48.1%	42.9%	48.6%	2	0.984
CG	365	41.4%	51.0%	40.8%	118	41.1%	42.9%	40.9%		
GG	92	10.4%	11.8%	10.3%	31	10.8%	14.3%	10.4%		
GPR44-1-RS545659										
AA	559	63.4%	68.6%	63.1%	184	64.1%	75.0%	62.9%	2	0.137
GA	265	30.0%	27.5%	30.2%	93	32.4%	21.4%	33.6%		
GG	58	6.6%	3.9%	6.7%	10	3.5%	3.6%	3.5%		
GPR44-2-RS533116										
AA	111	12.6%	11.8%	12.6%	48	16.7%	25.0%	15.8%	2	0.048
GA	383	43.4%	41.2%	43.6%	134	46.7%	46.4%	46.7%		
GG	388	44.0%	47.1%	43.8%	105	36.6%	28.6%	37.5%		
PTGFR-1-RS6686438										
GG	485	55.0%	60.8%	54.6%	146	50.9%	46.4%	51.4%	2	0.477
GT	326	37.0%	27.5%	37.5%	116	40.4%	42.9%	40.2%		
TT	71	8.0%	11.8%	7.8%	25	8.7%	10.7%	8.5%		
PTGFR-2-RS1328441										
AA	238	27.0%	27.5%	27.0%	60	20.9%	21.4%	20.8%	2	0.118
GA	423	48.0%	41.2%	48.4%	147	51.2%	60.7%	50.2%		
GG	221	25.1%	31.4%	24.7%	80	27.9%	17.9%	29.0%		
PTGFR-3-RS10782665										
GG	356	40.4%	41.2%	40.3%	114	39.7%	32.1%	40.5%	2	0.977
GT	381	43.2%	33.3%	43.8%	126	43.9%	50.0%	43.2%		
TT	145	16.4%	25.5%	15.9%	47	16.4%	17.9%	16.2%		
PTGES2-RS13283456										
CC	619	70.2%	80.4%	69.6%	195	67.9%	71.4%	67.6%	2	0.277
CT	230	26.1%	17.6%	26.6%	75	26.1%	21.4%	26.6%		
TT	33	3.7%	2.0%	3.9%	17	5.9%	7.1%	5.8%		
SULT1A1-RS9282861										
CC	464	52.6%	52.9%	52.6%	152	53.0%	50.0%	53.3%	2	0.001
TC	408	46.3%	43.1%	46.5%	122	42.5%	46.4%	42.1%		
TT	10	1.1%	3.9%	1.0%	13	4.5%	3.6%	4.6%		
CRABP2-RS12724719										
AA	29	3.3%	5.9%	3.1%	2	0.7%	0.0%	0.8%	2	<0.001
AG	278	31.5%	43.1%	30.8%	64	22.3%	17.9%	22.8%		
GG	575	65.2%	51.0%	66.1%	221	77.0%	82.1%	76.4%		
BTD-RS13078881										
CC	2	0.2%	0.0%	0.2%	0	0.0%	0.0%	0.0%	2	0.719
CG	60	6.8%	13.7%	6.4%	20	7.0%	7.1%	6.9%		
GG	820	93.0%	86.3%	93.4%	267	93.0%	92.9%	93.1%		
ACE-RS4343										
AA	213	24.1%	29.4%	23.8%	54	18.8%	25.0%	18.1%	2	0.081
AG	424	48.1%	33.3%	49.0%	158	55.1%	57.1%	54.8%		
GG	245	27.8%	37.3%	27.2%	75	26.1%	17.9%	27.0%		
IGF1R-RS2229765										
AA	149	16.9%	25.5%	16.4%	51	17.8%	17.9%	17.8%	2	0.888
AG	441	50.0%	33.3%	51.0%	145	50.5%	64.3%	49.0%		
GG	292	33.1%	41.2%	32.6%	91	31.7%	17.9%	33.2%		

**Table 2 medicina-59-01654-t002:** Model Summary.

Dimension	Cronbach’s Alpha	Variance Accounted for	
		Total (Eigenvalue)	% of Variance
1	0.359	1.506	9.416
2	0.303	1.396	8.726
Total	0.699	2.903	18.142

**Table 3 medicina-59-01654-t003:** Variance Accounted For.

	Centroid Coordinates			Total (Vector Coordinates)		
	Dimension		Mean	Dimension		Total
	1	2		1	2	
rs6198	0.016	0.079	0.047	0.016	0.079	0.094
rs2470152	0.015	0.007	0.011	0.015	0.006	0.021
rs39848	0.012	0.002	0.007	0.011	0.001	0.012
rs523349	0.003	0.003	0.003	0.003	0.003	0.005
rs545659	0.012	0.548	0.280	0.011	0.548	0.559
rs533116	0.003	0.582	0.292	0.002	0.582	0.584
rs6686438	0.359	0.003	0.181	0.359	0.001	0.360
rs1328441	0.433	0.010	0.221	0.432	0.010	0.442
rs10782665	0.549	0.000	0.274	0.549	0.000	0.549
rs13283456	0.006	0.017	0.011	0.006	0.016	0.022
rs9282861	0.046	0.010	0.028	0.045	0.010	0.055
rs12724719	0.011	0.000	0.006	0.011	0.000	0.011
rs13078881	0.003	0.001	0.002	0.003	0.001	0.004
rs4343	0.041	0.029	0.035	0.038	0.029	0.067
rs1800012	0.000	0.040	0.020	0.000	0.040	0.040
rs2229765	0.006	0.070	0.038	0.005	0.070	0.075
Active Total	1.513	1.400	1.457	1.506	1.396	2.903
% of Variance	9.458	8.752	9.105	9.416	8.726	18.142

**Table 4 medicina-59-01654-t004:** Component-Transformed Variables.

SNP	rs6198	rs2470152	rs39848	rs523349	rs545659	rs533116	rs6686438	rs1328441	rs10782665	rs13283456	rs9282861	rs12724719	rs13078881	rs4343	rs1800012	rs2229765
rs6198	1.000	0.005	−0.079	0.020	−0.078	−0.041	0.011	−0.002	0.053	0.00	0.017	0.032	−0.014	0.078	−0.007	−0.040
rs2470152	0.005	1.00	0.011	−0.018	−0.012	−0.016	−0.03	0.033	0.046	0.056	−0.002	0.026	0.005	0.043	0.007	0.021
s39848	−0.079	0.011	1.000	0.034	0.010	−0.036	−0.036	−0.010	−0.032	0.051	−0.022	−0.001	0.008	0.034	0.018	−0.04
s523349	0.020	−018	0.034	1.000	−0.020	0.040	−0.022	0.003	0.004	0.003	−0.033	0.019	0.001	−0.008	−0.021	0.007
s545659	−0.078	−0.012	0.010	−0.020	1.000	0.322	0.045	−0.057	0.016	0.010	0.014	−0.020	0.043	0.002	−0.030	0.055
rs533116	−0.041	−016	−0.036	0.040	0.322	1.000	−0.031	0.023	−0.026	−0.045	−0.048	0.034	−0.036	−0.040	−0.082	0.055
rs6686438	0.011	−0.003	−0.036	−0.022	0.045	−0.031	1.000	0.146	0.234	−0.014	0.033	−0.047	0.014	0.055	−0.002	0.008
s132B441	−0.002	0.033	−010	0.003	0.057	0.023	0.146	1.000	0.293	0.004	0.059	−0.014	−0.013	0.019	0.033	−0.028
s10782665	0.053	0.046	−0.032	0.004	0.016	−0.026	0.234	0.293	1.000	0.031	0.048	−0.025	0.022	0.033	−0.018	−0.010
s13283456	0.000	0.056	0.051	0.003	0.010	−0.045	−0.014	0.004	0.031	1.00	−0.004	0.036	0.041	0.078	−0.037	−0.051
rs9282861	0.017	−0.002	−0.022	−0.033	0.014	−0.048	0.033	0.059	0.048	−0.04	1.000	0.004	0.000	0.015	0.024	−0.030
s12724719	0.032	0.026	−0.01	0.019	−0.020	0.034	−0.047	−0.014	−0.025	0.036	0.004	1.000	−0.04	−0.014	0.034	0.013
rs1307881	−0.014	0.005	0.008	0.001	0.043	−0.036	0.014	−0.013	0.022	0.041	0.000	−0.04	1.000	0.042	−0.041	0.035
rs4343	0.078	0.043	0.034	−0.008	0.002	−0.040	0.055	0.019	0.033	0.078	0.015	−0.014	0.042	1.00	−0.009	−0.028
s180012	−0.007	0.007	0.018	−0.021	−0.030	−0.082	−0.002	0.033	−0.018	−0.037	0.024	0.034	−0.041	−0.009	1.000	0.002
rs2229765	−0.040	0.021	−0.004	0.007	0.055	0.055	0.008	−0.028	−0.010	−0.051	−0.030	0.013	0.035	−0.028	0.002	1.000
Dimension	1	2	3	4	5	6	7	8	9	10	11	12	13	14	15	16
Eigenvalue	1.506	1.396	1.170	1.096	1.069	1.044	1.037	0.990	0.976	0.942	0.907	0.880	0.855	0.821	0.673	0.638

**Table 5 medicina-59-01654-t005:** Genes and SNPs with significant differences in Brazil and Romania.

GENA	SNP	χ^2^	df	*p*
GR-ALPHA	RS6198	9.794	2	0.007
GPR44-2	RS533116	6.066	2	0.048
SULT1A1	RS9282861	13.368	2	0.001
CRABP2	RS12724719	16.210	2	<0.001

**Table 6 medicina-59-01654-t006:** Genes and SNPs with non-significant differences in Brazil and Romania.

GENA	SNP	χ^2^	df	*p*
CYP19A1	RS2470152	0.578	2	0.749
SRD5A1	RS39848	1.446	2	0.485
SRD5A2	RS523349	0.033	2	0.984
GPR44-1	RS545659	3.970	2	0.137
PTGFR-1	RS6686438	1.480	2	0.477
PTGFR-2	RS1328441	4.278	2	0.118
PTGFR-3	RS10782665	0.048	2	0.977
PTGES2	RS13283456	2.566	2	0.277
BTD	RS13078881	0.660	2	0.719
ACE	RS4343	5.031	2	0.081
IGF1R	RS2229765	0.238	2	0.888

## Data Availability

Available on request.

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
