# Peer review of "Epidemiologic Study of Gene Distribution in Romanian and Brazilian Patients with Non-Cicatricial Alopecia"

_medicina, 2023, doi:10.3390/medicina59091654_

Round 1

Reviewer 1 Report

Comments: Authors studied the ‘SNPs of 13 genes in patients from Romania and Brazil diagnosed with AA and AGA but manuscript requires attention on following comments and same needs to be addressed in the revised manuscript:

1)      Abstract: ‘The DNA samples were extracted from saliva using qPCR technique.’ Reframe sentence, its misleading/wrong. ‘Results: GR-alpha gene, GPR44-2 18 gene, SULT1A1 gene and CRABP2 gene were’ avoid repetition of gene word. ‘Minoxidil may be recommended in half’? no clarity in sentence?. Abstract (method and result) is misleading/confusing. Revise it appropriately.

2)      Introduction: “13 genes identified from previous genome-wide AA and AGA association studies.” What is the rationale of selecting genes, no reference provided to support the sentence?, methodology and data showed 15 genes, but 13 mentioned in text/title. Clarification required.

3)      Methodology: poorly explained,  any specific reason to isolate DNA from saliva, how PCR was performed, primers, PCR conditions, Taqman Technology? Gel picture of each SNPs/genotype should be provided as supplementary data.

4)      Grammar (e.g. tenses), Typos require special attention in the manuscript.

5)      Results: line 108, ‘In group 2 (and 882 Brazilian patients)’ ‘and’ not required, line 127, ‘prostaglandin analogue, Latanoprost.’, ‘and’ is required here, do proper proofreading of the manuscript to avoid such mistakes.

6)      Discussion: Authors should discuss their findings with findings of previously reported studies, is there any common SNP variation observed in other studied populations?

Line 343, ‘biotin (vitamin B7) might be involved in the development of AA. They reported hair regrowth in 12 of 19 patients with AA after biotin supplementation [40].’ Author want to say, ‘deficiency of biotin (vitamin B7) might be involved in the development of AA’??, sentence looks contradictory., line 345-46, no record of patients given who received what treatment, data interpretation is not possible as information is lacking.

7)      Conclusion: “GPR44 mRNA stability was found” No data mentioned to support the notion? Clarification required.

8)      6. Patents-???

Minor editing of English language required as mentioned in comments

Author Response

1)      Abstract: ‘The DNA samples were extracted from saliva using qPCR technique.’ Reframe sentence, its misleading/wrong. ‘Results: GR-alpha gene, GPR44-2 18 gene, SULT1A1 gene and CRABP2 gene were’ avoid repetition of gene word. ‘Minoxidil may be recommended in half’? no clarity in sentence?. Abstract (method and result) is misleading/confusing. Revise it appropriately. 

Thank you very much for taking the time to review. The abstract was revised as requested.

Modified version:

“We performed a retrospective study regarding the associations between AA and AGA and 45 tag SNPs of 15 genes in 287 Romanian and 882 Brazilian patients. The DNA samples were collected from oral mucosa using a swab. The SNPs were determined by the qPCR technique. Each genetic test displays the subject's genotype of the selected gene and the prediction of a successful treatment (e.g., genotype AA of the GR-alpha gene is related to a predisposition to normal sensibility to topical glucocorticoid and, therefore, glucocorticoids should be effective). Results: GR-alpha, GPR44-2, SULT1A1 and CRABP2 gene were statistically significantly different in Brazil compared to Romania. The SULT1A1 activity that predicts the response to minoxidil treatment, showed in our analysis that minoxidil is recommended in half of the AGA and AA cases.”

2)      Introduction: “13 genes identified from previous genome-wide AA and AGA association studies.” What is the rationale of selecting genes, no reference provided to support the sentence?, methodology and data showed 15 genes, but 13 mentioned in text/title. Clarification required.

The test analyzes the genes that were found to be most commonly associated with AA and AGA in the literature. Reference was introduced as requested (AA and AGA association studies “[8-12,18-20,23,35]”).

As this genetic test can be done in all types of alopecia, we sorted the genes, and two of them were initially removed then added again, because they were related to AGA and AA. I didn’t change the number from the manuscript. Thank you very much for being vigilant. 

3)      Methodology: poorly explained, any specific reason to isolate DNA from saliva, how PCR was performed, primers, PCR conditions, Taqman Technology? Gel picture of each SNPs/genotype should be provided as supplementary data.

  • Saliva:

The DNA was collected by using a swab from the oral mucosa. This method is used because it ensures enough DNA quantity and quality and is less invasive. It is a fast method with no risks for the patients during DNA harvest.

Modified version:

“The DNA samples were collected from oral mucosa using a swab. The method is non-invasive and ensures adequate DNA quantity and quality.

  • PCR:

This is a retrospective study that was done using the database of genetic information from Fagron Genomics. The PCR was not done during the study and I only had access to the retrospective data. All the genotyping assays were performed by qPCR Taqman Assays (Thermo Fisher Scientific) from genomic DNA previously collected from the patients. qPCR assays were designed and ordered from Thermo Fisher qPCR genotyping database.

Modified version:

“All the genotyping assays were performed by qPCR Taqman Assays (Thermo Fisher Scientific) from the genomic DNA collected from the patient.”

Fagron Genomics (Spain) provided all anonymized data “from their database for the current study.”

  • Gel picture:

We do not have gel picture because I only analyzed the retrospective data provided from Fagron from their database.

4)      Grammar (e.g. tenses), Typos require special attention in the manuscript.

Thank you very much. I have made the necessary changes.

5)      Results: line 108, ‘In group 2 (and 882 Brazilian patients)’ ‘and’ not required, line 127, ‘prostaglandin analogue, Latanoprost.’, ‘and’ is required here, do proper proofreading of the manuscript to avoid such mistakes.

Done. Thank you.

6)      Discussion: Authors should discuss their findings with findings of previously reported studies, is there any common SNP variation observed in other studied populations?

Line 343, ‘biotin (vitamin B7) might be involved in the development of AA. They reported hair regrowth in 12 of 19 patients with AA after biotin supplementation [40].’ Author want to say, ‘deficiency of biotin (vitamin B7) might be involved in the development of AA’??, sentence looks contradictory., line 345-46, no record of patients given who received what treatment, data interpretation is not possible as information is lacking.

Modified version:

“According to Georgala et al. [40], biotinidase deficiency can induce alopecia. The study showed that 12 out of 19 patients with significantly decreased biotinidase activity compared to controls, experienced clinical remission of AA after biotin supplementation. Moreover, Patel et al. [39] performed a literature analysis of 10 cases of alopecia—most patients with an inherited biotinidase enzyme deficiency or being treated with valproic acid. Eight patients experienced hair regrowth after biotin (vitamin B7) supplementation.

Our SNP analysis showed no indication of vitamin B supplementation in more than 90% of the cases of AA and AGA from both countries. Most of the evaluated subjects had normal biotinidase activity.”

7)      Conclusion: “GPR44 mRNA stability was found” No data mentioned to support the notion? Clarification required.

It was removed from the manuscript.

8)      6. Patents-??? 

“Patents” was removed.

Reviewer 2 Report

I have reviewed the paper by Paun et al.

Although the paper tries to link genetic analysis to the most appropriate treatment in the Introduction and in the conclusions, it is really just an epidemiological study comparing the distribution of genes associated with alopecia in two groups, one from Romania, and one from Brazil. This is well described in section 2 (Objectives), so the title should have been reflective of this aim.

Suggest removing the treatment paragraph in the Introduction, as it is not relevant to the study. A line mentioning that AGA and AA have different treatments should suffice.

I think it would be more useful to analyze the association of these genes with AGA and AA, rather than comparing between the two countries, which can still be done as a secondary aim. A PCA of these genes between both forms of alopecia would be a nice graph in this context.

Only then, starting a Discussion with “The possibility of linkages between SNPs in AA and AGA has only recently been investigated in a small number of Australian [8], European (Polish [5], UK and Germany [9])” would make sense.

The Discussion, does not discuss their findings, does not deal with differences amongst different populations. It is rather a minireview of the associations of these genes with different mechanisms of alopecia.

Some English editing required.

Author Response

  1. Although the paper tries to link genetic analysis to the most appropriate treatment in the Introduction and in the conclusions, it is really just an epidemiological study comparing the distribution of genes associated with alopecia in two groups, one from Romania, and one from Brazil. This is well described in section 2 (Objectives), so the title should have been reflective of this aim.

- Thank you for taking the time to review. The title was changed as requested.

“Epidemiologic study of gene distribution in Romanian and Brazilian patients with non-cicatricial alopecia”

  1. Suggest removing the treatment paragraph in the Introduction, as it is not relevant to the study. A line mentioning that AGA and AA have different treatments should suffice.

- Thank you for the suggestion. The last paragraph from the “Background” section was modified as required. As the article also refers to the treatments in AA and AGA, I listed only the treatments related to the genetic analysis but I can remove the paragraph.

  1. I think it would be more useful to analyze the association of these genes with AGA and AA, rather than comparing between the two countries, which can still be done as a secondary aim. A PCA of these genes between both forms of alopecia would be a nice graph in this context.

Only then, starting a Discussion with “The possibility of linkages between SNPs in AA and AGA has only recently been investigated in a small number of Australian [8], European (Polish [5], UK and Germany [9])” would make sense.

Thank you very much for the suggestions. We added a PCA in the manuscript in the statistical analysis section.

  1. The Discussion, does not discuss their findings, does not deal with differences amongst different populations. It is rather a minireview of the associations of these genes with different mechanisms of alopecia.

We tried to discuss all the analyzed genes and compare our results with the published ones. At the end of the paragraph, all concerned genes have our results and the association with the gene.

Round 2

Reviewer 2 Report

My comments have been addressed. I think the PCA addition was susbtantial to the paper.

Minor edits.